# Chemical Profile and Evaluation of the Antioxidant and Anti-Acetylcholinesterase Activities of *Annona squamosa* L. (Annonaceae) Extracts

**DOI:** 10.3390/foods10102343

**Published:** 2021-09-30

**Authors:** Débora Odília Duarte Leite, Cicera Janaine Camilo, Carla de Fatima Alves Nonato, Natália Kelly Gomes de Carvalho, Gerson Javier Torres Salazar, Selene Maia de Morais, José Galberto Martins da Costa

**Affiliations:** 1Northeast Biotechnology Network, Postgraduate Program in Biotechnology, State University of Ceará, Fortaleza 60740-000, CE, Brazil; biodeboraleite@yahoo.com.br (D.O.D.L.); selene.morais@uece.br (S.M.d.M.); 2Research Laboratory of Natural Products, Department of Biological Chemistry, Regional University of Cariri, Crato 63105-000, CE, Brazil; janainecamilo@hotmail.com (C.J.C.); carla.alves.bio@gmail.com (C.d.F.A.N.); nataliakellygc@gmail.com (N.K.G.d.C.); timotygertor@yahoo.com (G.J.T.S.)

**Keywords:** *Annona squamosa*, antioxidant, acetylcholinesterase, chemical prospection

## Abstract

This study presents the chemical profile of extracts from the pulp and seed of *Annona squamosa* L., as well as the evaluation of their antioxidant and acetylcholinesterase inhibition activities. In the chemical prospection, qualitative assays were performed, and the contents of total phenols, flavonoids, vitamin C, and carotenoids were quantified. For the compounds identification, analyses of the extracts were performed by liquid chromatography coupled to mass spectrometry. Antioxidant evaluation was performed using the DPPH, ABTS, Fe^3+^ reduction, 2-DR protection, and *β*-carotene protection methods. The assay for inhibition of acetylcholinesterase activity was determined using the method described by Ellman. The secondary metabolites identified were anthocyanidins, flavones, flavonols, and alkaloids. Phenol analysis showed a higher quantitative value of total phenols and flavonoids for the seed extract, and the vitamin C content was higher in the pulp extract. There was no significant difference in relation to the carotenoids quantification. The best results obtained for antioxidant activity, for both seed and pulp extracts, were with the ABTS method with IC_50_ of 0.14 ± 0.02 and 0.38 ± 0.02 mg/mL, respectively. Compared to *A. squamosa* seed extract, the pulp extract demonstrates higher AChE inhibitory activity with IC_50_ of 18.82 ± 0.17 µg/mL. *A. squamosa* is a nutritious food source. The continuity of the studies is fundamental to relate the consumption of this food and its effects on neurodegenerative diseases.

## 1. Introduction

Reactive Oxygen and Nitrogen Species (RONS), which are mostly free radicals, are highly reactive molecules and occur naturally in the body as a result of metabolism that maintain vital functions such as energy production, gene activation, and defense, among others [1]. Antioxidants are substances which can reduce or prevent the oxidation of substrates; they also occur naturally or can be acquired in food consumption. Generally, there should be a balance between RONS and antioxidants in the body; however, external factors such as pollutants, radiation, cigarette smoking, and pesticide use generate an excess of RONS that damage molecules such as DNA, RNA, proteins, and lipids, thus causing damage to the body [2]. This oxidative stress has as a main consequence lipid peroxidation, which is a process directly involved in the pathogenesis of diseases such as arthritis, asthma, dementia, Down syndrome, carcinoma, and degenerative diseases including Parkinson’s and Alzheimer’s [3,4].

Acetylcholinesterase (AChE) is an enzyme found in the central and peripheral nervous system (CNS). In the CNS it is involved in motor control, cognition, and memory; in the peripheral, it modulates the nerve impulses that control heartbeat, blood vessel dilation, and smooth muscle contraction [5]. Anticholinesterases prevent the degradation of acetylcholine by prolonging the effect of the nerve impulse, being important in controlling disorders that result in neural death such as Alzheimer’s disease [4]. Anticholinesterase agents can have reversible or irreversible action, and the drugs currently used have the disadvantage of having pronounced side effects, hence the need for new drugs found mainly from natural sources [5]. Antioxidant activity has been proven to be relevant in the treatment of Alzheimer’s disease as well, since natural antioxidant deficiency aggravated the disease in mice [6]. Therefore, products presenting antioxidant and acetylcholinesterase inhibition properties may represent a complementary alternative for the treatment of the disease [4].

*Annona squamosa* (Annonaceae) has an unknown origin and is cultivated in tropical countries, preferably in high temperature environments, such as the Brazilian semi-arid region [7]. Its fruits, usually consumed fresh, exhibit juicy, sugary, slightly acidic pulp [8]. Additionally, they are used in the manufacture of beverages (juices and soft drinks) and ice cream, among other products of the food industry, as well as being an important source of carbohydrates, proteins, vitamins, and minerals such as iron, calcium, and phosphorus. The vitamin C content of *A. squamosa* can exceed that of orange [9].

In addition to nutritional benefits, *A. squamosa* pulp can be used as antioxidant, antidiabetic, hepatoprotective, and antitumor agents, including activities related to alkaloids, fixed oils, tannins, and phenolic compounds present in its chemical composition [10]. The highest nutritional value is in the seed, with high protein content, high oleic, and linoleic acid value, thus making it a possible nutritional implementation in human food [11]. The aim of the present study was to chemically evaluate methanolic extracts of the pulp and seeds of *A. squamosa*, their antioxidant effect, and their potential for acetylcholinesterase inhibition.

## 2. Materials and Methods

### 2.1. Plant Material

The fruits were commercially acquired from the local market in Juazeiro do Norte, (Ceará, Brazil) in February 2019. The pulp was manually separated from the seed, submitted to lyophilization, and then kept in a desiccator until the use. The seeds were sanitized with sodium hypochlorite (5%), then washed with distilled water, kept in an oven (60 °C), after which they were ground to obtain powder.

### 2.2. Obtaining Extracts

The lyophilized pulp was extracted in a shaker incubator at 50 rpm for 72 h in a ratio of 1:1 (*m/v*) methanol. After this period, the material was filtered and concentrated under reduced pressure at 50 °C to obtain the crude extract (38%). The seed flour was extracted in a Soxhlet system with methanol until exhaustion, then submitted to a concentration process under reduced pressure at a temperature of 50 °C for the removal of the solvent, producing a crude extract with a yield of 8%.

### 2.3. Chemical Analysis

#### 2.3.1. Qualitative Tests

The chemical assays for detecting secondary metabolites were performed according to the methodology described by Matos [12], based on colorimetric readings or precipitate formation after addition of specific reagents.

#### 2.3.2. Quantification of Total Phenolic Content

Quantification of total phenols was determined by the Folin–Ciocalteu method [13]. For the general purpose, 25 µL of the diluted crude seed or pulp extract were added to 625 µL of 10% Folin–Ciocalteu reagent and 500 µL of 7.5% sodium carbonate. After incubation for 15 min in the dark at 45 °C, the absorbance was measured at 765 nm using the solvent as blank. Gallic acid was used as a standard for the calibration curve. The results were expressed as µg of gallic acid equivalent per mg extract. The analysis was performed in triplicate.

#### 2.3.3. Quantification of Total Flavonoids Content

Total flavonoid content was determined by the aluminum chloride colorimetric method [14]. Aliquots of 1160 µL of the diluted crude seed or pulp extract were added to 760 µL of solvent (methanol), 40 µL of 10% aluminum chloride and 40 µL of 0.1 M potassium acetate. After incubation for 30 min under the light at ambient temperature, the absorbance was measured at 415 nm using solvent as blank. Quercetin was used as standard for calibration curve. The results were expressed as µg of quercetin equivalent per g extract. The analysis was performed in triplicate.

#### 2.3.4. Total Vitamin C Determination

Ascorbic acid (AA) content was determined using the 2,6-dichlorophenol-indofenol titration method [15]. Portions of 10 g of pulp, or the flour from *A. squamosa* seeds, were mixed with 100 mL of a 2% oxalic acid solution. After filtration, a 10 mL aliquot of the mixture was titrated with 2,6-dichlorophenol-indofenol to a pink-colored end point. The dye factor was calculated by taking 5 mL of the standard ascorbic acid solution (0.05 mg/mL) added to 5 mL of 2% oxalic acid, then titrated as the dye solution. The dye factor was determined using the following formula.
$$\mathrm{Dye}\;\mathrm{factor}=\frac{0.5}{\mathrm{vol}.\mathrm{made}\;\mathrm{up}}$$

The amount of ascorbic acid (mg/100 g) was calculated as follows:$$\mathrm{Ascorbic}\;\mathrm{Acid}=\frac{\mathrm{vol}.\mathrm{made}\;\mathrm{up}\;\times \;\mathrm{dye}\;\mathrm{factor}\;\times \;\mathrm{total}\;\mathrm{vol}.\times 100}{\mathrm{Aliquot}\;\mathrm{used}\;\mathrm{for}\;\mathrm{titrate}\;\times \;\mathrm{wt}\;\mathrm{of}\;\mathrm{sample}}$$

#### 2.3.5. Determination of Total Carotenoids

Total carotenoid content was determined using the method adapted by Huang and colleagues [16], in which 1 g of pulp or the seed meal of *A. squamosa* was subjected to carotenoid extraction with 25 mL of acetone:ethanol (1:1) containing 200 mg/L BHT. The mixture was filtered and washed with 60 mL of the extraction solvent, then diluted to a volume of 100 mL, 50mL of hexane was added, stirred, and left to stand for 15 min. Twenty-five milliliters of double distilled water was added, the mixture was stirred, and then left resting for another 30 min. The sample absorbance was measured in a spectrophotometer set at 470 nm. The carotenoids were quantified as *β*-carotene equivalent using a standard curve of this compound, i.e., mg of *β*-carotene/g dry weight.

#### 2.3.6. LC-MS Analysis

The extracts were analyzed by Shimadzu HPLC using an analytical chromatographic column C18 (Kromasil—250 mm × 4.6 mm × 5 µm), coupled to a mass spectrometer (Ion-Trap AmazonX, Bruker), with Electrospray Ionization (ESI). To perform the analysis, the sample was solubilized in methanol (1 mg/mL), with subsequent filtration through PVDF (Polyvinylidene Fluoride) filters with a mesh size of 0.45 μm. The chromatographic method that was developed used the solvents methanol (solvent B) of chromatographic grade and ultrapure water type I (Milli-Q), acidified with formic acid (0.1% *v*/*v*) (solvent A), with concentration gradient analysis (5 to 100% of B in 95 min.). The injection volume was 10 μL and flow rate was 0.6 mL/min. In the mass spectrometer, the samples were subjected to sequential fragmentation in MS3. The parameters used were: capillary 4.5 kV, end plate offset 500 V, nebulizer gas 35 psi, dry gas (N_2_) with a flow rate of 8 mL/min and temperature of 300 °C. The samples were analyzed in negative ionization mode, and the identification of the compounds was based on the fragments (MS/MS) reported in the MassBank database.

### 2.4. Antioxidant Activity of Extracts

#### 2.4.1. DPPH (2,2-Diphenyl-1-picrylhydrazyl) Method

The test was performed using the method described by Rufino et al. [17]. The reaction mean consisted of 150 μL of the crude seed or pulp extract diluted at concentrations ranging from 125 to 2000 µg/mL and 150 μL of DPPH solution (0.15 mM). Ascorbic acid and methanol were used for positive and negative control, respectively. The reaction was carried out for 30 min in the dark. The reading was performed with a spectrophotometer at 515 nm. The results were calculated according to the equation:AA% = 100−{[(Abs_sample_−Abs_blank_)/Abs_negative control_] × 100}(1)
where, AA% refers to the percentage of antioxidant activity, and Abs is absorbances. The analysis was performed in triplicate.

#### 2.4.2. ABTS (2,2′-Azino-bis(3-ethylbenzothiazoline-6-sulphonic Acid)) Method

The radical capture was measured by the method proposed by Rufino et al. [18]. Initially, the ABTS^•+^ solution was obtained from the reaction of ABTS solution (7 mM) with sodium persulfate solution (140 mM) that was kept for 16 h in the shelter of light. The diluted crude seed or pulp extract concentrations ranged from 125 to 2000 µg/mL. Under dark conditions, a 30 µL aliquot of each concentration was transferred to test tubes with 3.0 mL of the ABTS^•+^ radical. The reading was performed in a spectrophotometer at 734 nm after 6 min of reaction of the mixture. Pure ABTS^•+^ radical was used as negative control, methyl alcohol as blank, and ascorbic acid as positive control. The results were calculated according to Equation (1). The analysis was performed in triplicate. 

#### 2.4.3. Ferric-Reducing Antioxidant Power Method

The extract’s ability to reduce Fe^3+^ was determined by the o-phenanthroline assay, according to the method of Minotti and Aust [19], with a few modifications. Aliquots of the crude seed or pulp extracts diluted to the different concentrations were mixed with Fe^2+^ ion in vitro using a 1200 μM FeSO_4_ solution, and with Fe^3+^ ion in vitro using a 1200 μM FeCl_3_ solution, separately in a dark, refrigerated environment. After 2.5 min of reaction, 50 μL of the mixtures were transferred to 96-well plates containing Milli-Q water, 0.1 M Tris-HCl pH 7.4, and 300 μM *o*-phe, obtaining final concentrations of the extracts from 125 to 2000 µg/mL, 100 μM of Fe^2+^ ion, and 100 μM of Fe^3+^ ion. The absorbance was measured in a spectrophotometer at 510 nm simultaneously with controls where aliquots of extracts were replaced by Milli-Q water. The results of Fe^+3^ reducing power were expressed as a percentage compared to Fe^+2^ controls according to the following equation:(2)$$\mathrm{Reducing}\;\mathrm{Power}\;(\%)=\frac{(\left(\mathrm{Abs}.\mathrm{Ext}.\mathrm{Fe}+3-\mathrm{Abs}.\mathrm{Ext}.\mathrm{white}\right)-\left(\mathrm{Abs}.\mathrm{Cont}.\;\mathrm{Fe}+3-\mathrm{Abs}.\mathrm{white}.\mathrm{Fe}+3\right)\times 100}{\mathrm{Abs}.\mathrm{Cont}.\mathrm{Fe}+2-\mathrm{Abs}.\mathrm{white}\;\mathrm{Fe}+2\;}$$
where, Abs._Ext. Fe+3_ is the absorbance of the extract with *o*-phe and Fe^+3^; Abs._Ext. white_. is the absorbance of the extract without *o*-phe and without Fe^+2^; Abs._Cont. Fe+3_ is the absorbance of *o*-phe with Fe^+3^ without extract; Abs._white Fe+3_ is the absorbance of Fe^+3^ without extract and without *o*-phe; Abs._Cont. Fe+2_ is the absorbance of *o*-phe with Fe^+2^ without extract; and Abs._bran Fe+2_ is the absorbance of Fe^+2^ without extract and without *o*-phe. The analysis was performed in triplicate.

#### 2.4.4. Oxidative Degradation of 2-Deoxyribose (2-DR) Method

The method used was adapted from Puntel et al. [20]. The methanolic solution of the extracts was prepared at concentrations from 125 to 2000 µg/mL. For the test, 450 μL potassium phosphate buffer (7.5 mM), 150 μL deoxyribose (1.5 mM), 240 μL H_2_O_2_ (0.8 mM), 240 μL FeSO_4_ (80 µM), and 320 μL distilled water were added to 100 μL of the crude seed or pulp extracts diluted at the different concentrations. For the blank, the same mixture was prepared in the absence of deoxyribose. All samples were incubated for 60 min at 37 °C. Thereafter, they received 750 μL of trichloroacetic acid (2.8%) and 750 μL of thiobarbituric acid (0.8%), respectively, and were re-incubated for 20 min in a heated bath at 100 °C. The reading was performed in a spectrophotometer at 532 nm. The results were calculated according to Equation (1). The analysis was performed in triplicate with ascorbic acid as positive control.

#### 2.4.5. Co-Oxidation of *β*-Carotene/Linoleic Acid Method

The method used was an adaptation of the one proposed by Rufino et al. [21]. The *β*-carotene solution (1 mL) was prepared by dissolving 20 mg of *β*-carotene in 1 mL of chloroform; an aliquot (25 μL) of this solution was placed in a round bottom flask containing 20 μL of linoleic acid and 265 μL of the emulsifier Tween 40. After removing the chloroform, in a rotary evaporator at 40 °C, distilled water (previously saturated with oxygen for 30 min) was added until absorbance between 0.6 and 0.7 at 470 nm was obtained. 250 μL of this emulsion were transferred to micro-plate wells containing 20 μL of the crude seed or pulp extract at concentrations from 125 to 2000 µg/mL. Afterwards, the tubes were placed in a Bain-marie at 45 °C for 120 min and the absorbance measured at 470 nm initially and after 120 min. Vitamin C was used as a standard. The blank was prepared in the absence of the emulsion. The antioxidant activity was expressed according to Equation (1).

### 2.5. Micro-Plate Assay for Inhibition of Acetylcholinesterase

The assay for inhibition of acetylcholinesterase activity was determined using the method described by Ellman et al. [22]. The following solutions were added in 96-well plates: 25 µL acetylcholine iodide (15 mM), 125 µL 5,5-dithiobis-[2-nitrobenzoic acid] in Tris/HCl solution (50 nM, pH = 8, with 0.1 M NaCl and 0.02 M MgCl_2_.6H_2_O, Ellman’s reagent), 50 µL of Tris/HCl solution containing bovine serum albumin, 25 µL of the crude seed or pulp extract at different concentrations. The absorbance was measured for 30 s at 405 nm, then 25 µL of the enzyme acetylcholinesterase was added, and the absorbance was read every minute during 30 min. The standard used as a control was physostigmine, and the percentage of acetylcholinesterase inhibition was calculated using equation:Inhibition (%) = 100−[Abs_sample_−Abs_water_/Abs_control_] × 100(3)
where Abs_sample_ corresponds to the absorbance of the sample, Abs_wate_ corresponds to the absorbance referring to the coloring of the extract, and Abs_control_ represents the absorbance of the total enzyme activity without the presence of the extract. The analysis was performed in duplicate.

## 3. Results

### 3.1. Chemical Analyses

The classes of secondary metabolites found in both the pulp and seed extracts were anthocyanidins, flavones, and flavonols. The seed of *A. squamosa* revealed the presence of alkaloids. Phytochemical assays performed by other researchers using chemical reactions with extracts from the aerial parts of *A. squamosa* identified alkaloids, quinones, saponins, resins, essential oils, reducing sugars, flavonoids, carbohydrate-phenols, and tannins [23]. A phytochemical screening of *A. squamosa* leaves by Kalidindi and colleagues [24] using extracts from different solvents revealed the presence of higher amounts of metabolite classes using methanol as the solvent, with the presence of alkaloids, glycosides, flavonoids, tannins, carbohydrates, phenols, and saponins identified in the extract.

The results show significantly higher total phenols value for seeds 32.53 ± 3.6 µg GAE/mg Ext. (Table 1). Studies with the leaf extract presented results for total phenols in the range of 54.75 to 352.0 mg/100 g of dry-powder samples [25]. The results presented by Huang and collaborators [16] used dehydrated pulp and seed, obtaining for the pulp a total phenol content equal to 1.5 ± 0.03 g gallic acid/100 g dry weight, and for the seed 0.3 ± 0.01 g gallic acid/100 g dry weight. Other studies with the fresh fruit had total phenol results of 233 ± 23.8 mg gallic acid/100 g fresh weight [26], 81.7 ± 4.0 mg gallic acid/100 g fresh weight [27], 747 ± 15.1 mg gallic acid/100 g fresh weight [28] and 207.60 ±17.85 mg gallic acid/100 g fresh weight [29]. 

The result of total flavonoids was also higher in the seed extract, 893.3 ± 11.55 µg QE/g Ext (Table 1). It is possible to find flavonoid values for fresh pulp equal to 170.6 ± 5.9 mg CE/100 g [28] and 200.92 ± 3.83 mg QE/100 g [29].

According to the tests analyzed in this study and presented in Table 1, total flavonoids represent 2.7% of the total phenols value for the seed extract, and 11.2% for the pulp extract, suggesting the presence of other phenolic groups. Phenolic acids were identified in the fresh pulp of *A. squamosa* by Baskara and collaborators [30], consisting of about 16 free phenolic acids, 15 bound, and 13 esterified. In the seeds, flavonoids such as Rutin, Isoquercitrin, Quercitrin, and Quercetin have been identified in addition to gallic acid [31,32]. Chemical studies of the seeds have also identified the presence of flavonoids, tannins, leucoanthocyanins, triterpenes, unsaturated sterols, polyphenols, and polysaccharides [33]. The variety of phenolic compounds of *A. squamosa* is scientifically known, supporting the quantification of the current study.

In this study, the vitamin C content is significantly higher in the pulp than it is in the seed extract, as shown in Table 1. For the quantification of carotenoids there was no significant difference between the methanolic extracts of the pulp and seeds of *A. squamosa* (Table 1). The variation of carotenoids is mainly related to temperature and light intensity since these substances have chromophores responsible for the absorption of visible light spectrum; it is possible that during processing and storage there is a change in the constitution of carotenoids [34]. Studies by Huang and collaborators [16] found the vitamin C and carotenoids content in *A. squamosa* seeds was 10.8 ± 0.56 µmol/g DW and 6.6 ± 0.37 µg *β*-carotene/g DW, respectively, in the pulp the vitamin C content was 8.1 ± 0.16 µmol/g DW and carotenoids 6.4 ± 0.29 µg *β*-carotene/g DW; the values found for the seed in the present studies are higher. Some other works also quantified vitamin C and carotenoids in the pulp of *A. squamosa* with results of 29.6 ± 0.9 µmol/g dry weight, 21.43 ± 4.99 µmol/g dry weight, 31.22 ± 0.15 mg/100 mL for vitamin C and 2.97 ± 0.64 mg *β*-carotene/g dry weight for carotenoids [11,27,28].

The chromatographic analysis of the methanolic extract of *A. squamosa* pulp, working under the conditions employed and through the similarity of mass-to-charge ratio and fragments (MS2) with data described in the MassBank database, allows us to suggest the presence of oleic acid (1, Rt = 67 min) at *m/z* = 281.25. In the methanolic extract of the seeds it was possible to suggest the presence of three compounds: linoleic acid (2, Rt = 66 min) at *m/z* = 279.59, palmitic acid (3, Rt = 68 min) at *m/z* = 281.27, and stearic acid (4, Rt = 71.5 min) at *m/z* = 283.72. These compounds have already been reported to be present in *A. squamosa* seeds according to the literature [11,31,33]. Chemical structures of the identified compounds are presented in Figure 1.

### 3.2. Antioxidant Activity

The best results regarding the evaluation of the antioxidant potential were in the reduction of the ABTS radical, in which the extract of *A. squamosa* seeds more efficiently showed an IC_50_ of 0.14 ± 0.02294 mg/mL. The other results are presented in Table 2. 

Studies show the antioxidant potential of phenolic compounds as being considered the probable contributors to antioxidant activity. The evidence is given by the positive correlation between the content of phenols and antioxidant capacity [35,36]. Among phenolic compounds, flavonoids represent the main group providing antioxidant properties in plant extracts [28]. The presence of hydroxyls in phenolic compound molecules is mainly responsible for their antioxidant activity, playing a role of donating hydrogen, removing free radicals, and reactive species [37].

Taking as reference the phenolic acids, the most frequent phenolic compounds in *A. squamosa* pulp, identified by Baskaran and colleagues [30], ferulic acid has an electron donating methoxyl group, and the presence of a hydroxyl in the ortho position provides stability to the phenyl radical. Caffeic acid, which has a second hydroxyl in the para position, has a potentiated antioxidant effect compared to ferulic acid. A double bond present in molecules derived from cinnamic acid, such as synaptic, ferulic, and *p*-coumaric acids, stabilizes the radical formed by the antioxidant process through the paired electron shift resonance [38].

These chemical characteristics present in phenolic compounds identified in *A. squamosa* contribute to its antioxidant potential.

Vitamin C is considered a powerful antioxidant in vitro; it represents a good reducing agent whose generated radical is of a low-reactivity type, and can also eliminate non-radical reactive species; the antioxidant effects in vivo are still uncertain, but the levels of vitamin C present in the body are compatible with its antioxidant potential in vitro. In the presence of transition metals, they can exhibit pro-oxidant effect, producing hydrogen peroxide and forming hydroxyls [39]. Carotenoids, found in the pulp and seed of *A. squamosa*, also contribute to the antioxidant activity, since their mechanism of action involves the suppression of superoxide O_2_^−•^, a highly reactive form which can generate cell damage, and also inactivate molecules in a state of excitation, especially those arising from photosensitive reactions. High concentrations can lead to pro-oxidant action, accelerating the oxidation rate [40].

Vikas and collaborators [41] performed tests with extracts from the seeds of *A. squamosa* in order to evaluate the free radical scavenging capacity, obtaining IC_50_ of 28.8 µg/mL for the DPPH radical and IC_50_ of 80 µg/mL for the ABTS radical with methanolic extract. By using chloroform as the solvent, it was possible to obtain more significant values for the reduction of the ABTS radical, with IC_50_ of 73.25 µg/mL. The authors also evaluated the ability to reduce iron using the FRAP reagent; methanolic extract was the most efficient with IC_50_ of 67 µg/mL, and the remaining results were 70.5, 74, and 74 µg/mL for extraction with ethyl acetate, chloroform, and petroleum ether, respectively. By using chloroform as solvent, it was possible to obtain more significant values for the reduction of the ABTS radical, with IC_50_ of 73.25 µg/mL. Such discrepant results in the present study may be related to the concentration of the reagents and the environment concentration of the plant material with different climatic conditions. Studies of the antioxidant potential with *A. squamosa* seeds are scarcely available due to the presence of acetogenins in its chemical composition, an extremely toxic class of compound widely used in research for the treatment of tumor cells.

For Silva and Serasa [28], the pulp of *A. squamosa* presents a great antioxidant potential, against the DPPH radical the methanolic extract obtained a result of 27.37 mg AEAC/g of fresh fruit, and 46.75 µmol FeSO4/g for the reduction of iron using the FRAP reagent. Huchin and collaborators [29] prepared pulp extract with acetone/water/acetic acid (70:29.5:0.5 *v*/*v*/*v*) using vitamin C and trolox as standard; results for DPPH were 77.12 VCEAC mg/100g fruit and 357.88 TEAC µg/100g fruit, for ABTS radical, 191.43 mg VCEAC/100g and 655.7 µg TEAC/100g. The authors suggest a significant potential for the development of new products with functional properties.

### 3.3. Inhibition of Acetylcholinesterase

According to the results obtained, the extract of *A. squamosa* pulp showed better AChE inhibitory activity than the extract of seeds. This result, however, is lower than the positive control physostigmine that presents IC50 of 1.15 ± 0.05 µg/mL. The decrease in enzyme inhibition over time suggests a reversible action for both the seed extract and the pulp extract, as shown in Figure 2 and Figure 3. Further studies are needed to prove the reversible nature of these samples, such as analysis over a longer period of time and in vivo tests that can corroborate the in vitro findings. At the highest concentration of 200 µg/mL, the inhibition was higher, reaching 74.4% for the pulp extract, and 73% for the seed extract, with the IC_50_ in the first minute being 22.31 ± 0.17 µg/mL for the seed and 18.82 ± 0.17 µg/mL for the pulp. There is a lack of studies related to AChE inhibition with this species, which makes comparison with published data difficult. Sultana [42] performed anticholinesterase tests with ethanolic extract of *A. squamosa* seeds and concluded that it was inactive for AChE inhibition with an inhibition rate of 0.5%. There is general agreement among nutritionists that vitamin C is positively related to a decrease in the incidence of neurodegenerative diseases [43]. The area with noradrenaline-producing neurons are regions affected by Alzheimer’s; this neurotransmitter is responsible for interest, attention, stress, and other reactions to the environment, as well as quantitative changes that can lead to lack of sleep, memory loss, and inflammation [44]. Vitamin C is a cofactor of dopamine *β*-monooxygenase, whose function is the synthesis of noradrenaline [43]. Seeds and pulp of *A. squamosa* have considerable value of vitamin C, which is important for the prevention and treatment of neurodegenerative diseases. *A. squamosa* is also rich in fatty acid. Studies indicate that it acts on the G-protein coupled receptor of the cerebral cortex, promoting neurogenesis, inhibiting neural apoptosis, protecting nerves, and reducing brain damage; furthermore, it is directly linked to Alzheimer’s disease by releasing insulin according to the amount of glucose, which is compromised due to the pathology [45].

Alzheimer’s disease is characterized by an accumulation of plaques consisting of *β*-amyloid peptides, which contribute negatively to mitochondrial function; this impairment in the respiratory chain leads to oxidative stress, accumulating reactive oxygen species that add up to deleterious effects in the disease. Therefore, diets rich in antioxidant substances may slow or even prevent neurological deterioration caused by oxidative stress. Plants that exhibit positive effects for cognitive disorders, such as antioxidant and anti-acetylcholinesterase activity, represent a potential candidate for clinical and nutritional use [46,47]. Detailed studies are needed to attest to the efficiency of *Annona squamosa* in the prevention and even treatment of neurodegenerative diseases, such as Alzheimer’s disease.

## 4. Conclusions

The extract of the seeds and pulp of *A. squamosa* presents dose-dependent inhibitory activity for AChE with a decrease in enzyme inhibition over time; detailed studies are needed to prove its reversible nature, if confirmed, it favors the use of this plant as a complement in the treatment of neurodegenerative diseases, especially at the beginning of the manifestation, since the presence of vitamin C is directly linked to the production of noradrenaline, an important neurotransmitter in memory. Carotenoids consumption also aids in the prevention of related degenerative diseases. Fatty acids, found in great quantities in this fruit, also act in the treatment of diseases that affect the nervous system, regulating in a controlled way insulin release, and avoiding hypoglycemia in the brain, mainly in the case of Alzheimer’s disease. The moderate antioxidant activity of these extracts represents prevention to the aggravation of these diseases. As a source of vitamin C and phenolic compounds, *A. squamosa* can be considered a nutritious food source, allowing it to be prescribed in the diet of patients with Alzheimer’s and/or other neurodegenerative diseases. Healthy patients who are predisposed to these diseases can also add the fruit to their diet as a preventive measure.

## Figures and Tables

**Figure 1 foods-10-02343-f001:**
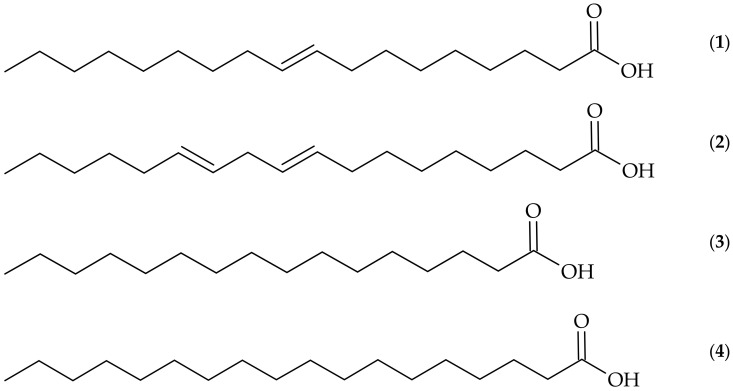
Structures of the constituents identified in the extracts of *Annona squamosa*: (**1**) oleic acid; (**2**) linoleic acid; (**3**) palmitic acid; (**4**) stearic acid.

**Figure 2 foods-10-02343-f002:**
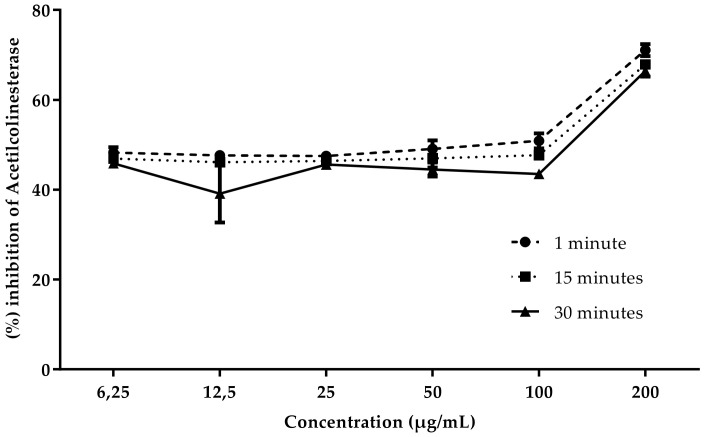
Concentration-dependent AChE inhibitory activity of methanolic extract of *A. squamosa* seeds.

**Figure 3 foods-10-02343-f003:**
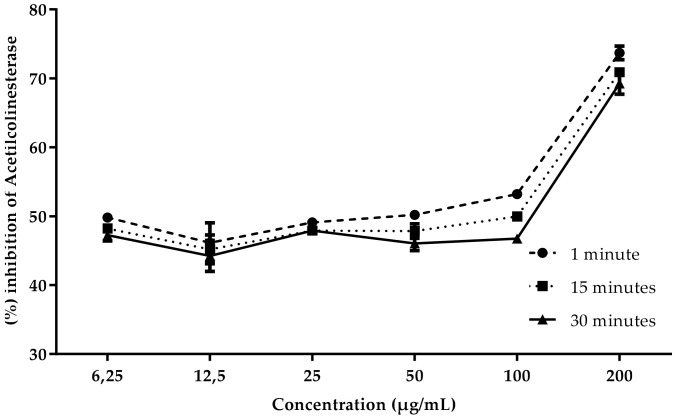
Concentration-dependent AChE inhibitory activity of methanolic extract of *A. squamosa* pulp.

**Table 1 foods-10-02343-t001:** Total phenols and total flavonoids quantification from *A. squamosa* seeds and pulp.

	Total Phenolics (µg GAE/mg Ext.)	Total Flavonoids (µg QE/g Ext.)	Vitamin C (mg AA/100 g)	Caratenoids (µg of *β*-Carotene/10 mg)
Seeds	32.53 ± 2.13 ^a^	893.30 ± 6.66 ^a^	0.57 ± 0.07 ^a^	0.45 ± 0.01 ^a^
Pulp	2.20 ± 0.09 ^b^	246.60 ± 23.33 ^b^	1.01 ± 0.08 ^b^	0.38 ± 0.03 ^a^

These results are expressed as mean ± SEM (*n* = 3). Values followed by different letters in the same column (a or b) differ statistically by *t*-test, *p* < 0.05.

**Table 2 foods-10-02343-t002:** Antioxidant activity of *A. squamosa* seed and pulp extracts.

	IC_50_ (mg/mL)				
	DPPH	ABTS	Fe^3+^ Reduction	2-DR Protection	*β*-Carotene Protection
Seeds	0.36 ± 0.02	0.14 ± 0.02	0.57 ± 0.01	0.41 ± 0.019	0.16 ± 0.03
Pulp	0.83 ± 0.02	0.38 ± 0.02	0.74 ± 0.05	0.43 ± 0.16	1.36 ± 0.02
Vitamin C *	0.011 ± 0.04	0.004 ± 0.01	0.031 ± 0.05	0.042 ± 0.04	0.95 ± 0.18

* Positive control. Data are expressed as mean ± SD (*n* = 4).

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
