# Peer review of "Chemical Profile and Evaluation of the Antioxidant and Anti-Acetylcholinesterase Activities of Annona squamosa L. (Annonaceae) Extracts"

_foods, 2021, doi:10.3390/foods10102343_

Round 1

Reviewer 1 Report

In general, the article evaluates the antioxidant capacity of a product native to tropical countries and that its consumption could have an effect on health. However, the study applied, most of the tests carried out have already been studied by other authors and makes it not very original.

Some observations or concerns are:

Abstract

The phrase "A. squamosa can be considered as a nutritious food source 
that can be prescribed in the diet of patients with Alzheimer's and/or other neurodegenerative diseases" is quite affirmative, is there proof of this from human studies?

Materials and methods

In the methodology: "The pulp was manually separated from the seed, subjected to freeze-drying and then 
 freeze-dried, and then preserved in a desiccator until use. The seeds were sanitized with sodium hypochlorite (5 %), then washed with distilled water, kept in an oven (60 76°C), after which they were ground to obtain the powder".

The freeze-drying method and the high temperatures would not put the analyzed compounds at risk of loss?

Results

In the sentence "According to the results obtained, the pulp extract of A. squamosa showed a better AChA inhibitory activity than the pulp extract of A. squamosa.  AChE inhibitory activity than the seed extract. This result, however, is inferior to that of the positive control  c physostigmine which has an IC50 of 1.15 ± 0.05 µg/mL". 

According to the above, are the results obtained sufficient in quantity to treat a disease such as Alzheimer's? or what quantity of this fruit should be consumed to have a biological effect?

Conclusion

You should write as in the analysis of the results, what more study is needed to explain the reversible action of AChE inhibitory activity 
of AChE and the decrease of enzymatic inhibition over time.

Author Response

Answers to the comments_Reviewer 1

1) Abstract

The phrase "A. squamosa can be considered as a nutritious food source 
that can be prescribed in the diet of patients with Alzheimer's and/or other neurodegenerative diseases" is quite affirmative, is there proof of this from human studies?

Replaced

  1. squamosa is a nutritious food source. The continuity of the studies is fundamental to relate the consumption of this food and its effects on neurodegenerative diseases.

2) Materials and methods

In the methodology: "The pulp was manually separated from the seed, subjected to freeze-drying and then 
freeze-dried, and then preserved in a desiccator until use. The seeds were sanitized with sodium hypochlorite (5 %), then washed with distilled water, kept in an oven (60 °C), after which they were ground to obtain the powder".

The freeze-drying method and the high temperatures would not put the analyzed compounds at risk of loss?

Secondary metabolites, such as those of the classes reported here, are resistant to elevated temperatures. The upper temperature limit (60 °C) allows extraction of compounds and does not cause thermal damage to the material, especially when combined with exposure time. Sultana (2007), referenced in the article, also makes use of similar methodology.

3) Results

In the sentence "According to the results obtained, the pulp extract of A. squamosa showed a better AChA inhibitory activity than the pulp extract of A. squamosa.  AChE inhibitory activity than the seed extract. This result, however, is inferior to that of the positive control  c physostigmine which has an IC50 of 1.15 ± 0.05 µg/mL". 

According to the above, are the results obtained sufficient in quantity to treat a disease such as Alzheimer's? or what quantity of this fruit should be consumed to have a biological effect?

Added paragraph stating the need for detailed studies. The results found represent important data to start this research.

4) Conclusion

You should write as in the analysis of the results, what more study is needed to explain the reversible action of AChE inhibitory activity 
of AChE and the decrease of enzymatic inhibition over time.

corrections performed

Reviewer 2 Report

The authors investigate the antioxidant power and acetylcholinesterase inhibition activity of Annona squamosa pulp and seed extracts. They claim to have found, through HPLC-MS, anthocyanins, flavones and flavonols and some alkaloids but do not report any details on the molecules. Some picture of the chromatograms could help the reader. The article focuses only on the total content of phenols, total flavonoids, vitamin C and carotenoids both in the pulp and in the seeds, furthermore the antioxidant activity is evaluated by different methods. Finally, they discuss acetylcholinesterase inhibition.

The authors report the methods used in the experiments; however, they do not clarify how they obtained the values shown in Table 1 from the chromatogram, calibration lines figures would be welcome.

Discussion of the results needs improvement because there is a long list of references from which the parameter values are taken but they are poorly compared with those obtained in the experiments. All these values could be summarized in tabular form so as to make reading and comparison more immediate.

Some sentences lead the reader to think that the reported values derive from experiments while referring to literature items.

Sometimes different units are used for values from the same parameter which is a confusing factor for the reader.

Hereafter a list of flaw in addition to the previous comments:

  • Page 3 lines 106: “760 µL of solvent” specify which solvent.
  • Page 3 lines 117: “dye factor” indicate how calculated this parameter.
  • Page 3 lines 128: “the extraction solvent, there after diluted” perhaps “the extraction solvent, then after dilution”.
  • Page 4 lines 161-162: “From which, AA% refers to the percentage of antioxidant activity, and Abs, absorbances.” better “where AA% refers to the percentage of antioxidant activity and Abs to the absorbances.”.
  • Page 4 equation 2: It is advisable to use a symbol to indicate the Ferric-reducing antioxidant power percentage, not simply the symbol “(%) =”.
  • Page 4 lines 188: “From which” better “where”.
  • Page 5 lines 214: “until absorbance between 0.6 and 0.7 nm” unit of measurement inconsistent with the parameter.
  • Page 5 section “Chemical analyses”: it is not easy to understand, especially in lines 241-250, when talking about literature data or values drawn from the experiments. Table 1, furthermore, is poorly commented in the text.
  • Page 4 lines 234: “From which” better “where”.
  • Page 6 lines 277: “carotenoid content, 10.8 ± 0.56 µmol/ g dry weight” The unit µmol/g is not coherent with the unit of measurement in the header of the column of Vitamin C in table 1. I think there is a mistyping in table 1.
  • Page 6 Table 1: The units of measurement used in the columns “Vitamin C (mg AA/100 g)” and “Caratenoids (µg of β-carotene/10mg)” are different from those used in the text in lines 277-280. I suggest always using the same unit for the values coming from the same parameter, this in order to simplify the reading. Check the unit of measurement coherence all over the paper.
  • Page 7 Figure 1: Revise the caption, explain the meaning of the numbers and brackets in the figure.
  • Page 7 section “Antioxidant Activity”: make it more evident when talking about data from literature and when discussing experimental results.

Author Response

Answers to the comments – Reviewers 2

1) The authors investigate the antioxidant power and acetylcholinesterase inhibition activity of Annona squamosa pulp and seed extracts. They claim to have found, through HPLC-MS, anthocyanins, flavones and flavonols and some alkaloids but do not report any details on the molecules. Some picture of the chromatograms could help the reader. The article focuses only on the total content of phenols, total flavonoids, vitamin C and carotenoids both in the pulp and in the seeds, furthermore the antioxidant activity is evaluated by different methods. Finally, they discuss acetylcholinesterase inhibition.

The data refers to qualitative analysis as described in the methodology

2) The authors report the methods used in the experiments; however, they do not clarify how they obtained the values shown in Table 1 from the chromatogram, calibration lines figures would be welcome.

The data refers to qualitative analysis as described in the methodology

3) Discussion of the results needs improvement because there is a long list of references from which the parameter values are taken but they are poorly compared with those obtained in the experiments. All these values could be summarized in tabular form so as to make reading and comparison more immediate.

Corrections performed

4) Some sentences lead the reader to think that the reported values derive from experiments while referring to literature items.

Corrections performed

5) Sometimes different units are used for values from the same parameter which is a confusing factor for the reader.

Corrections performed

6) Hereafter a list of flaw in addition to the previous comments:

  • Page 3 lines 106: “760 µL of solvent” specify which solvent. Corrections performed
  • Page 3 lines 117: “dye factor” indicate how calculated this parameter. Corrections performed
  • Page 3 lines 128: “the extraction solvent, there after diluted” perhaps “the extraction solvent, then after dilution”. Corrections performed
  • Page 4 lines 161-162: “From which, AA% refers to the percentage of antioxidant activity, and Abs, absorbances.” better “where AA% refers to the percentage of antioxidant activity and Abs to the absorbances.”. Corrections performed
  • Page 4 equation 2: It is advisable to use a symbol to indicate the Ferric-reducing antioxidant power percentage, not simply the symbol “(%) =”. Corrections performed
  • Page 4 lines 188: “From which” better “where”. Corrections performed
  • Page 5 lines 214: “until absorbance between 0.6 and 0.7 nm” unit of measurement inconsistent with the parameter. Corrections performed
  • Page 5 section “Chemical analyses”: it is not easy to understand, especially in lines 241-250, when talking about literature data or values drawn from the experiments. Table 1, furthermore, is poorly commented in the text. Corrections performed
  • Page 4 lines 234: “From which” better “where”. Corrections performed
  • Page 6 lines 277: “carotenoid content, 10.8 ± 0.56 µmol/ g dry weight” The unit µmol/g is not coherent with the unit of measurement in the header of the column of Vitamin C in table 1. I think there is a mistyping in table 1. Refers to data from the literature. Corrections performed
  • Page 6 Table 1: The units of measurement used in the columns “Vitamin C (mg AA/100 g)” and “Caratenoids (µg of β-carotene/10mg)” are different from those used in the text in lines 277-280. I suggest always using the same unit for the values coming from the same parameter, this in order to simplify the reading. Check the unit of measurement coherence all over the paper. Refers to data from the literature. Corrections performed
  • Page 7 Figure 1: Revise the caption, explain the meaning of the numbers and brackets in the figure. A new structure was presented to make it easier to read
  • Page 7 section “Antioxidant Activity”: make it more evident when talking about data from literature and when discussing experimental results. Corrections performed

Round 2

Reviewer 1 Report

The new manuscript submitted by the authors contains the corrections suggested in revision 1.

Considering the above, and that no further experiments were necessary.